# Single-Cell RNA Sequencing Reveals Heterogeneity and Functional Diversity of Lymphatic Endothelial Cells

**DOI:** 10.3390/ijms222111976

**Published:** 2021-11-05

**Authors:** Hannah den Braanker, Astrid C. van Stigt, Marc R. Kok, Erik Lubberts, Radjesh J. Bisoendial

**Affiliations:** 1Department of Clinical Immunology and Rheumatology, Maasstad Hospital, Maasstadweg 21, 3079 DZ Rotterdam, The Netherlands; BraankerH@maasstadziekenhuis.nll (H.d.B.); KokMR@maasstadziekenhuis.nl (M.R.K.); 2Department of Rheumatology, Erasmus University Medical Center, Doctor Molewaterplein 40, 3015 GD Rotterdam, The Netherlands; a.vanstigt@erasmusmc.nl (A.C.v.S.); e.lubberts@erasmusmc.nl (E.L.); 3Department of Immunology, Erasmus University Medical Center, Doctor Molewaterplein 40, 3015 GD Rotterdam, The Netherlands

**Keywords:** lymphatics, lymphatic endothelial cells, single cell RNA-sequencing, lymph nodes, inflammation

## Abstract

Lymphatic endothelial cells (LECs) line the lymphatic vasculature and play a central role in the immune response. LECs have abilities to regulate immune transport, to promote immune cell survival, and to cross present antigens to dendritic cells. Single-cell RNA sequencing (scRNA) technology has accelerated new discoveries in the field of lymphatic vascular biology. This review will summarize these new findings in regard to embryonic development, LEC heterogeneity with associated functional diversity, and interactions with other cells. Depending on the organ, location in the lymphatic vascular tree, and micro-environmental conditions, LECs feature unique properties and tasks. Furthermore, adjacent stromal cells need the support of LECs for fulfilling their tasks in the immune response, such as immune cell transport and antigen presentation. Although aberrant lymphatic vasculature has been observed in a number of chronic inflammatory diseases, the knowledge on LEC heterogeneity and functional diversity in these diseases is limited. Combining scRNA sequencing data with imaging and more in-depth functional experiments will advance our knowledge of LECs in health and disease. Building the case, the LEC could be put forward as a new therapeutic target in chronic inflammatory diseases, counterweighting the current immune-cell focused therapies.

## 1. Introduction

The lymphatic vasculature plays an essential role in many physiological and pathological processes in the human body. Although the origin of the lymphatic vasculature is closely connected to the vascular system, it has unique tasks and properties. Amongst these are the regulation of interstitial fluid balance, clearance of tissue-derived debris, and lipids and immune cell transport. Thus far, tissue fluid homeostasis has been the best studied function of the lymphatic vasculature, but novel roles are being increasingly discovered [1,2,3]. Tissue inflammation or injury, featured by increased interstitial fluid and enhanced influx of immune cells, requires expansion of the lymphatic vasculature to facilitate transport of tissue debris, leukocytes, and antigens (Ag), either soluble or carried by Ag-presenting cells, to the lymph node (LN). These events lead to an efficient immune response, and, ultimately, resolution of inflammation. In the last 20 years, identification of lymphatic markers, new experimental in vivo models, and 2D and 3D culture assays with lymphatic endothelial cells (LECs) have advanced our knowledge on lymphatic vascular biology [4]. This has important implications for a wide spectrum of disease areas, such as wound healing, obesity, cancer, and chronic inflammation and autoimmunity [5,6,7,8]. Changes in lymphatic vasculature, such as dilated lymphatic vessels or decreased amount of lymphatic vessels, are also widely observed in chronic (rheumatic) inflammatory diseases, such as psoriasis, rheumatoid arthritis (RA), systemic sclerosis, and inflammatory bowel disease (IBD) [9,10,11,12].

The recently developed technique of single-cell RNA sequencing (scRNA-seq) is widely used to uncover the identity and heterogeneity of cell subpopulations, and it has, either alone or combined with other state-of-the-art techniques, increased our knowledge on new functions of LECs in health and disease. The scRNA-seq technique evolved from sequencing only a handful of cells to currently sequencing more than 100,000 cells per sample [13]. ScRNA-seq protocols usually consist of the following steps: single cell capture, single cell lysis, reverse transcription, preamplification, library preparation, and sequencing. Laser capture microdissection, fluorescence-activated cell sorting (FACS) in microtiter plates, microfluidics, and microdroplets can be used to capture single cells. Techniques can differ in the number of cells that can be included, sequencing depth, and costs [14]. High dimensional data analysis has been applied to obtain biologically meaningful results. Frequently used strategies include dimensionality reduction methods, such as principal component analysis (PCA), t-distributed stochastic neighbor embedding (t-SNE), and uniform manifold approximation and projection (UMAP), followed by clustering algorithms, such as hierarchical clustering or k-means clustering [15]. Other promising approaches, such as trajectory inference and cell–cell interaction plots, intend to predict the origin of cells and interactions with other cells [16,17].

Although the number of scRNA-seq studies in the lymphatic vascular field are limited, a few exciting new discoveries were made in the last four years. This review will discuss the recent obtained knowledge about the lymphatic system with scRNA-seq techniques. For that matter, we will discuss recent insights in embryonic development, LEC heterogeneity with related functional diversity, LEC-stromal cell interactions, and transcriptomic changes found in LECs in inflammation. 

## 2. Single-Cell Transcriptomic Insides in Embryonic Development

Studies on embryonic development can often explain complex pathological processes, as has been demonstrated by Burchill et al. using scRNA-seq of liver LECs in an inflammatory environment [18].

### 2.1. Peripheral Lymphatic Vessel Formation

In mammals, both the origin of LECs lining the lymphatic vessels and the underlying molecular mechanisms and proteins involved in the embryonic development are not entirely clear. Historically, LECs were thought to exclusively originate from embryonic venous cells [19,20,21,22]. However, more recent studies also found that specific lymphatic vascular beds in the skin, the gut, and the heart can bear LECs from non-venous origins [23,24,25,26]. Molecular heterogeneity between LECs isolated from different organs also supports the possibility of multiple sources for LECs [27,28,29]. Unfortunately, the scRNA-seq technique has not yet been used to study the origin of LECs in the different lymphatic vascular beds.

The underlying molecular mechanisms and contributors of the venous-to-lymphatic transition have been best studied. This process starts with the transcription factor Nuclear Receptor Subfamily2 Group F Member 2 (NR2F2), also called COUP-TFII, cooperating with SRY-Box Transcription factor 18 (SOX18) to activate Prospero homeobox protein 1 (PROX1) expression [30]. PROX1 is the master transcription factor in LECs and drives expression of key markers, such as vascular endothelial growth factor receptor 3 (VEGFR-3), integrin-9alpha (adhesion molecule), podoplanin (cell migration), and chemokine (C-C motif) ligand 21 (CCL21) (chemokine) [20,31,32,33]. Downregulation of *Prox1* and its target genes result in loss of LEC identity and function, as was observed in a scRNA-seq mouse liver study [18]. According to this study, LEC identity and function could be regained by exposing them to VEGF-C. VEGF-C is one of the five members of the VEGF family of angiogenic cytokines that signals through VEGFR-3, which, in turn, signals via the growth factor-regulated phosphoinositide 3-kinase (PI3K)–AKT signaling network or the Mitogen-activated protein kinase (MAPK)/extracellular signal-regulated kinases (ERK) pathways to regulate embryonic development of LECs and LEC identity, survival, proliferation, and migration in adulthood [34].

Based on data of scRNA-seq studies, new genes have been suggested to play a role in embryonic lymphatic development. A recent scRNA-seq study in a murine newborn lung model predicted, aside from *Prox-1, Sox18, and Nr2f2*, for example, *homeobox d 8 (Hoxd8), Maf, T-box 1 (Tbx1),* and *ETS Transcription Factor 3 (Elk3)* to be involved in LEC development [35]. *Hoxd8* is a homeodomain transcription factor and can induce and maintain *Prox-1* expression [30]. *Maf* is a transcription factor upregulated by VEGF-C/VEGFR3 signaling that plays a role in developmental lymphangiogenesis [36]. *Tbx1* belongs to the T-box transcription factors, which are mostly involved in developmental processes, and *Tbx1* mouse mutants have severe problems with pharyngeal and cardiovascular development. Interestingly, conditional deletion of *Tbx1* in mice also causes widespread lymphangiogenesis defects due to its role in regulating VEGFR3 expression [37]. *Elk3* is an Erythroblast Transformation Specific (ETS) binding domain transcription factor and has not yet been studied in the context of embryonic development. However, *ELK3*-expressing LECs promote breast cancer growth and migration of cancer cells [38]. As published earlier, NR2F2 can form heterodimers with PROX-1 to activate Notch signaling in LECs important for lymphangiogenesis [39]. Other unknown protein-coding genes found in the scRNA-seq in the newborn lung study include, for example, *Profilin 1 (Pfn1)* and AE Binding Protein 1(*Aebp1)* [35]. Future studies into these genes might confirm roles for these factors in embryonic development.

Strikingly, a recent study combining an scRNA-seq approach validated by experimental mouse models discovered that the protein Folliculin (FLCN), long known as a tumor suppressor protein, plays a major role in controlling venous-to-lymphatic cell transition. FLCN prevents expression of *Prox-1* in venous endothelial cells (VECs) by binding of the *Prox-1* regulatory element, the transcription factor E3 (*Tfe3*). FLCN deficiency results in disrupted lymphatic and venous development and dilatation of vessels [40]. Figure 1A shows an overview of venous-to-lymphatic transdifferentiation. Many processes in embryonic development driven by these transcription factors have been implicated in post-natal lymphangiogenesis during inflammation. However, knowledge on aberrant expression of many of these transcription factors is lacking.

### 2.2. Lymph Node Development

The embryonic development of the lymph node (LN) is a step-by-step process and requires close interaction between mesenchymal cells and lymphoid tissue inducer (LTi) cells (Figure 1B). First, mesenchymal cells recruit LTi cells by expressing chemokine (C-X-C motif) ligand 13 (CXCL13) (Figure 1B, step 1). LTi cells cluster together and are identified by expression of chemokine (C-X-C motif) receptor 5 (CXCR5), the receptor for CXCL13, CD45, CD4, and CD127 (Figure 1B, step 2). LTi cells also express receptor activator nuclear factor κ B (RANK) and its ligand, RANKL, which induces high amounts of lymphotoxin α1β2 (LT1β2) [41,42,43]. LT1β2 activates the lymphtoxin beta receptor (LTβR) on mesenchymal cells. Subsequently, LTβR induces expression of cell adhesion molecules, such as intracellular adhesion molecule 1 (ICAM-1), and chemokines, such as CXCL13, CCL21, and CCL19, by mesenchymal cells. This initiates a positive feedback loop that attracts more LTi cells to the site (Figure 1B, step 3). The mesenchymal cells differentiate eventually in the stromal cells found in the lymph node. Whether LTi cells differentiate into LECs or, at their turn, attract LECs to the LN site is currently unknown [44]. Developed LNs consist of a subcapular sinus (SCS), which is connected to the afferent lymphatic vessels, and they collect migrating immune cells from draining tissues, including T- and B-lymphocytes and dendritic cells (DCs) [8]. Resident CD169+ macrophages and DCs in the SCS of the LN provide Ags, assembled from captured pathogens and molecules/proteins, to passing lymphocytes [45,46,47]. This is an essential interaction for educating lymphocytes towards proper immune responses. Lymphocytes are activated in LN parenchyma and travel through the cortical sinus (CS) and medullary sinus (MS) before they exit the LN into the peripheral blood [48].

Trajectory inference analysis of lymph node LECs predicted that LECs from the ceiling of the SCS and LECs from collecting lymphatic vessels have a shared origin, and they give rise to lymphatic valve cells (Figure 1B, step 4). LECs lining the floor of the SCS and in the MS also seem to have the same origin. No functional experiments have been performed yet to confirm these predicted developmental paths of LN LECs. Better insights in LN development are needed and will also help to further determine how tertiary lymphoid organs (TLOs) develop. TLOs, induced by LTi cells, are frequently found in chronically inflamed tissue, where they regulate inflammation and consist of many stromal cells, also found in lymph nodes [49]. Thus far, TLOs have not been investigated with scRNA-seq methodology yet.

## 3. The Heterogenous Family of LECs

### 3.1. Lymph Node LEC Subpopulations

LECs mark the outline of SCS, CS, and MS in the LN, along with other stromal cells, such as fibroblastic reticular cells and perivascular cells [50]. ScRNA-seq studies have revealed transcriptomic differences between LEC populations lining these different parts of the LN, and they have identified subpopulations that delineate the floor of SCS (floor LECs), the ceiling of the SCS (ceiling LECs), the medullary sinus (medullary LECs), and the cortical sinus (cortical LECs) [51,52,53]. Figure 2 gives an overview of the different subpopulations, their spatial location, markers, and proposed functions. Differences between mice and human LN subpopulations may be species-specific, but they are more likely the result of technical differences related to the scRNA-seq technique or sequencing depth.

#### 3.1.1. Floor LECs

LECs lining the floor of the SCS were identified in both mice and human LNs. Immune cells that arrive at the SCS, seem to report to floor LECs to further migrate to the LN parenchyma. Firstly, scRNA-seq data in mice and human LNs revealed expression of many adhesion and migration molecules, such as *integrin alpha 2b*, *Madcam-1, Glycam-1,* and *Lyve-1*. This confirms the role of the SCS in receiving lymph-derived cells from draining tissues, and to conduct immune cells to other parts of the LN. Since *Lyve-1* was only expressed in murine floor LECs and not in human floor LECs, also confirmed by immunohistochemical stainings, migratory cells like DCs may use molecules other than Lyve-1 in the human LN to migrate over floor LECs [54].

Floor LECs also express chemokines and costimulatory and coinhibitory molecules associated with T cells. *Ccl20 (or MIP-3A),* the sole ligand for the chemokine receptor CCR6, is highly expressed in floor LECs. CCR6 is expressed on many different immune cells, including IL-17A-producing memory T helper cells [55], also called Th17 cells, and gamma delta (γδ) T cells, which are potential drivers of many chronic inflammatory diseases [56]. CCR6 expression was also found on IL-17A+ innate-like lymphocytes (ILCs) in the mouse LN SCS. These ILCs were essential for innate responses against lymph-borne pathogens [57].

Floor LECs might also have a role in immunological tolerance. They express molecules such as *TNFRSF9*, a costimulatory molecule for T cells, and *PDL1* and *IFNGR*, coinhibitory signals for CD8+ T cells [52,53]. In addition, Floor LECs express neutrophil-attracting chemokines, such as *CXCL5* and *CXCL1* [53]. However, immunohistochemical stainings did not reveal adherence of neutrophils on floor LECs. Although trajectory inference also predicted that floor LECs play an essential role in the immune response, more studies are needed to uncover the full functionality of this unique LEC subpopulation.

#### 3.1.2. Ceiling LECs

The ceiling LECs of the SCS that are adjacent to the LN capsule seem to affect transport of immune cells in a different way. In contrast to floor LECs, scRNA-seq data revealed expression of the chemokine scavenging receptor ACKR4 and Ackr3 [51,52,53]. Chemokine scavenging receptors create chemokine gradients to guide immune cell transport, and their expression is unique to ceiling LECs. Other molecules expressed by ceiling LECs are the immunoregulatory molecule CD73 and Cd36, a lipid transporter. Interestingly, CD73 is an important enzyme for adenosine production. CD73 knock-out mice exhibited increased proinflammatory gene expression in LECs of LN, resulting in increased recruitment of inflammatory DCs upon immunological challenge [58]. Thus, CD73+ LECs have anti-inflammatory properties. Aberrant CD73 gene expression has not yet been identified in chronic inflammatory diseases, but it could be highly interesting. Afferent lymph also frequently contains fatty acids, such as low-density lipoproteins (LDLs). CD36 is a fatty acid transporter and is involved in LDL transport. Histological analysis confirmed CD36 expression in ceiling LECs and found that ceiling LECs selectively take up acetylated LDL, whereas floor LECs take up oxidized LDL [52].

#### 3.1.3. Cortical LECs

Cortical LECs delineate the cortical sinus and direct immune cells to the medulla. Immune cells in the cortex originate either from the afferent lymph and migrate over the floor LECs of the SCS or enter the LN via the high endothelial venules (HEVs). HEVs recruit naïve and memory lymphocytes from the blood. Under physiological conditions, lymphocytes rapidly enter the cortical sinus and egress from the LN towards the circulation. In the inflamed LN, lymphocytes are hampered to enter the cortical sinus and retained in the cortex to facilitate more T and B cell interactions in germinal centers of the cortex [59].

Using a scRNA-seq strategy, Fujimoto et al. identified a cluster of cortical LECs that specifically regulate lymphocyte egress from the cortex to the cortical sinus. ScRNA-seq data, together with immunohistochemical staining, identified Anxa-2, Lyve-1, pentraxin 3 (Ptx3), potassium inwardly rectifying channel subfamily J member 8 (Kcnj8), and inter-alpha-trypsin inhibitor heavy chain 5 (Itih5) that mark these cortical LECs. [52] Interestingly, Ptx3 belongs to the pentraxin family and, amongst other functions, mediates the host response to several micro-organisms that determine the course of inflammation [60]. Since the LN also has a function in filtering tissue-derived lymph, the expression of molecules, such as Ptx3 by cortical LECs, may contribute to alerting the LN of ongoing inflammation. Additionally, cortical LECs expressing these genes were in close proximity to the HEVs. Using labeled murine splenocyte experiments, Fujimoto et al. confirmed with in vivo models that these cortical sinuses, lined by cortical LECs, are reached by lymphocytes via the LN cortex. In the human LN, cortical LECs expressed LYVE-1 and Macrophage Receptor With Collagenous Structure (MARCO), but they were transcriptomically indifferent from medullary LECs, which could be due to a limited sequencing depth [52]. MARCO is a scavenger receptor class-A protein, mostly expressed on macrophages, and it mediates binding and uptake of environmental particles, such as ferric oxide or particles of bacteria and viruses [61]. Future studies need to elucidate the role of cortical LECs in the human LN and their critical role in regulating lymphocyte egress from the LN.

#### 3.1.4. Medullary LECs

Medullary LECs line the medullary sinus and transport cells and lymph to the medulla and efferent lymphatic vessel. Many B cell precursors migrate to the medulla to mature into plasma cells and secrete antibodies in the lymph. In the human LN, medullary LECs share their transcriptomic profiles with cortical LECs and partly with floor LECs. Amongst many cell adhesion molecules, Medullary LECs express neutrophil attractants, such as *CXCL1*, together with C-type lectins, such as *CLEC4G*, *CLEC4M*, and *CD209* [53]. Confirmed by immunohistochemistry, neutrophils were found to mostly adhere to medullary LECs. Ex vivo assays blocking CD209 and its partner, Lewis X, showed that neutrophils were unable to adhere to the medullary LECs, confirming the role of CD209 in neutrophil adherence in medullary LECs. Neutrophil adherence in the medullary sinus can be important for clearance of lymph-borne pathogens, such as skin-derived *Staphylococcus aureus* [53]. Furthermore, medullary LECs in human and mice LN express MRC1 and MARCO. These molecules can function as bacteria-binding molecules and recognize bacterial polysaccharides. Together with the neutrophil data, these results suggest a key role for medullary LECs in filtering lymph for pathogens.

### 3.2. Peripheral LEC Subpopulations

In the periphery, lymphatic capillaries resorb tissue fluid, antigens, and growth factors and enable transport of migration immune cells by means of their oak-leaf shape and button-like junctions and a discontinuous basal membrane by which leukocytes or macromolecules can pass (Figure 2) [47,62,63]. Smooth muscle cells control further propulsion of lymph to the collecting lymphatic vessels. Intraluminal valves ensure unidirectional flow of lymph in the collecting vessels [64].

ScRNA-seq data of peripheral LECs is limited. Takeda et al. sequenced a few LN surrounding collecting lymphatic vessels and lymphatic capillaries. The LECs from the afferent collecting vessels shared the expression of *CD73* and *CAV1* with ceiling LECs of the SCS. *PDPN, LYVE-1,* and *CCL21* were highly expressed on lymphatic capillaries, and *CLDN11* was found as a marker for valve LECs [53]. It is unclear if LEC subpopulations within capillaries or collecting vessels exist. Furthermore, if LECs populations differ between organs is unknown. However, in a large scRNA-seq study of endothelial cells in mouse organs, LECs from different organs clustered together and did not seem transcriptionally different [65]. Another large mouse endothelial scRNA-seq study also observed minimal differences between LECs from different tissues, except for LECs from the gut [66].

Multiple scRNA-seq studies sequencing whole organs or tissues have found an LEC population identified by the expression of LYVE-1, PROX-1, PDPN, VEGFR3, and CCL21 [66,67,68,69,70]. In these studies, a few additional molecules were found to be expressed by LECs. Murine LECs in multiple organs express other molecules, such as *Thy1, Mmrn1,* and *Esam,* mostly involved in extra cellular matrix remodeling and cell migration [66]. In the cornea, LECs were found to additionally express many major histocompatibility complex (MHC) I genes and angiogenic privilege factors [67]. Lung LECs highly express *SEMD3D,* a semaphorin that is known to guide entry of DCs into lymphatics [69,71]. In the mouse liver, LECs express *Mmrn1* but also *Tbx1,* a molecule involved in lymphangiogenesis, and *Ahnak2* [37]. *Ahnak2* is a large protein, and it has been found in endothelial cells in blood–tissue barriers [72]. The LECs in the mouse liver also highly expressed *Cd36*, indicating their role in LDL transport [18]. LECs were also found in a scRNA-seq study of human placentas. These LECs were specifically located in the maternal portion of the placenta, the decidua basalis. Pathways for cell migration and adhesion were highly enriched in these cells, indicating a role in regulating immune cell inflex in the decidua basalis [73]. The LEC clusters in most studies only consist of at most 1% of the sequenced cells. More in-depth scRNA-seq studies will hopefully provide more insights into LECs from different organs. Functional studies into some of the found expressed markers are also lacking and could potentially uncover new functions of LECs.

## 4. LECs Interact with and Support Other Stromal Cells

Throughout the whole body, LECs are in close proximity to other stromal cells.

ScRNA-seq data has been widely used to uncover a few of the interactions and their molecular mechanisms by LECs with other stromal cells, such as microglia, fibroblasts, and perivascular cells.

### 4.1. Interactions of LECs in the Meninges

Meningeal lymphatic drainage of macromolecules is important for normal brain function. Disruption of normal lymphatic flow can induce cognitive impairment in mice and plays a role in the development of Alzheimer’s disease (AD). Impairing meningeal lymphatics revealed significant changes in single-cell sequenced microglia and blood endothelial cells of the blood–brain barrier. Microglia showed transcriptomic changes in cytokine production, antigen processing, and antigen presentation and in pathways attracting, regulating migration of, and activating leukocytes and myeloid cells. Blood endothelial cells (BECs) showed changes in genes involved in interferon-γ signaling, cytokine-mediated signaling pathways, and monocyte chemotaxis. Modulating the lymphatic vasculature by administering VEGF-C ameliorated the disease severity in the AD mouse model [74]. These data suggest close interactions between meningeal LECs, microglia, and BECs and a key role for LECs in coordinating immune responses in the brain. Unfortunately, no studies have investigated meningeal LECs in, for example, multiple sclerosis yet. 

### 4.2. Lacteal LEC-Fibroblast Interactions

Lacteals are specialized lymphatic units in the villi of the small intestines which absorb dietary fats and fat-soluble vitamins. In large scRNA-seq studies that looked at endothelial cells of different mouse organs, lacteal LECs were classified as a specific subpopulation of LECs [66]. The LECs in these lacteals are constantly renewing and are supported by stromal cells, such as perivascular cells, fibroblasts, and blood capillaries. An in-depth study combining scRNA-seq and mouse models to study the structure and function of lacteals found that lacteal LECs interact with a specific subset of fibroblasts in the lacteal. This subpopulation of Platelet-derived growth factor receptor beta (PDGFRβ^+^) fibroblasts secretes VEGF-C to support lacteal LEC function and maintain discontinuous button-like junctions of lacteal LECs needed for dietary fat absorption. The VEGF-C production in these fibroblasts was regulated by the Hippo signaling pathway proteins YAP and TAZ [75]. In diseases such as Crohn’s disease, where a higher intake of dietary fats is a known risk factor and aberrant lymphatic vasculature is observed, future research into lacteal LECs and associated fibroblast subpopulations could be interesting in the context of pathogenesis and therapeutic targets [76].

### 4.3. LECs and Perivascular Cells

Perivascular cells, including vascular smooth muscle cells (SMCs), perivascular macrophages, and pericytes, maintain homeostasis and support many functions of the blood and lymphatic vasculature. Interactions between LECs and perivascular cells are not widely studied. For blood endothelium, the interaction with perivascular cells is essential for angiogenesis, maintenance, and remodeling [77]. Perivascular cells can have paradoxical roles in diseases, such as atherosclerosis and heart disease. They are needed for angiogenesis after ischemia, but they also produce a lot of extracellular matrix that results in fibrosis in the tissue [78]. SMCs, specifically those expressing *Klf4*, have been especially studied in the context of diseases, because they are mostly found in atherosclerotic plaques [79]. A recent scRNA-seq study investigated SMCs in adipose tissue in the context of diet-induced obesity. The authors found multiple subsets of perivascular cells. A SMC subpopulation expressing *Klf4* lined not only blood capillaries, but also the lymphatic capillaries. Surprisingly, knocking out this *Klf4^+^* SMC in a mouse model for diet-induced obesity increased the number of LECs in adipose tissue. This increase of LECs resulted in improved glucose tolerance [80]. These data suggest that altered lymphangiogenesis and lymph drainage could contribute to the pathogenesis of obesity. However, more studies need to confirm LEC and perivascular cell crosstalk in health and disease and the impact of perivascular cells on lymphangiogenesis and remodeling.

## 5. Transcriptomic Changes in LECs in Inflammation

In inflamed tissue, interstitial fluid accumulates, and interstitial pressure rises. The increased interstitial pressure is thought to mechanically induce the VEGFR3-VEGFC pathway to promote lymphangiogenesis [81]. Lymphangiogenesis in inflamed tissue results in more and expanded lymphatic vessels equipped to transport more lymph with tissue debris and immune cells to the LN. In LN, VEGF-C is needed to expand the LN under inflammatory conditions. Follicular B-cells have been reported to be major producers of VEGF [82,83]. The production of VEGF-C in follicular B-cells is increased upon exposure to common inflammatory mediators, like TNF, which signals through the NF-κβ signaling pathway [22,84]. TNF can also activate the NF-κβ pathway in LECs, enhancing VEGFR-3 expression and again chemokine and cytokine expression [22,54,84,85,86,87]. Only a few scRNA-seq studies have investigated inflammatory changes in LECs in periphery and LN. Figure 3 shows an overview of transcriptional changes observed in LECs from inflamed tissue. 

### 5.1. Inflammatory Changes in LN LECs

Sibler et al. scRNA sequenced inflamed auricular (draining) LNs from mice, challenged with imiquimod. Imiquimod, an imidazoquinoline amine, is an immune response stimulator that binds to toll-like receptor 7 and 8, activating various immune cells and their cytokine production. Applying imiquimod to mouse ears induces a psoriasis-like inflammation [88]. The floor LECs displayed the most transcriptional changes in the inflamed LN, whereas only modest transcriptional changes were found in ceiling LECs and medullary LECs. Interestingly, the transcriptional changes in inflamed LN subsets did only minorly overlap, again underlining the heterogeneity and diverse function of LN subsets. Floor LECs significantly upregulated *Ccl20,* indicating their function in attracting CCR6^+^ expressing cells, such as CCR6^+^ IL-17-producing cells, important in the imiquimod mice model (Figure 3A) [89]. Furthermore, floor LECs also upregulated *CD200* and *Anxa2* in the inflamed LN. CD200 receptors are expressed by DCs and macrophages and activated T cells. CD200–CD200R interactions can regulate macrophage function. Interestingly, in systemic lupus erythematosus (SLE), blocking CD200R on CD4^+^ T cells reduced the Th17 percentage in vitro. In vitro enhancing CD200 signaling also enhanced regulatory T cell percentages [90]. However, the role of CD200 expression by LECs need to be further investigated. *Anxa2* is a molecule involved in many different cell processes. In the cortical sinuses in close proximity to HEVs and supporting fast egress of lymphocytes, *Anxa2* expression was also found [52]. These results might indicate a role for *Anxa2* in the LN in lymphocyte migration, which becomes more important in the inflamed LN in the floor LECs.

### 5.2. Inflammatory Changes in Peripheral LECs

It is known that peripheral LECs upregulate expression of chemokines, such as CCL21, in inflammation [5]. A recent scRNA-seq study of mouse endothelial cells revealed that CCL21 expression is coupled to expression of the sphingosine 1-phosphate (S1P) /S1P receptor 1 (S1PR1) signaling pathway in LECs. S1P expression by LECs controls lymphocyte egress to the LN [59]. Ablating S1P/S1PR1 signaling also upregulated many other cytokine-signaling paths in LECs, including *Il33, Il7,* and *Irf8*. Lymphangiogenesis related genes were also upregulated in the S1P/S1PR1 knock out mouse model [91]. These data suggest a broader range of function for S1P/S1PR1 signaling in LECs during inflammation, but further studies are needed.

Burchill et al. studied inflammatory changes in mouse liver LECs after a challenge with oxidized LDL. Oxidized LDL has been implicated in inflammatory processes in many organs. In a murine model of non-alcoholic steatohepatitis (NASH), ScRNA-sequencing of LECs in the diseased liver revealed numerous transcriptional changes. Most strikingly, LEC identity markers were downregulated, such as *Pdpn* and *Prox-1*, also found in the immunohistochemical staining. They also found an upregulation of *Cd36*, the fatty acid transporter, also found in ceiling LECs in the LN [92]. CD36 knock-out mice show lymphatic dysfunction and increased risk for obesity and insulin resistance. [93] In the human NASH liver, scRNA-seq data also confirmed downregulation of key genes in LECs by oxidized LDL [18]. The effects of oxidized LDL on LECs are displayed in Figure 3B.

Over the years, it has been increasingly acknowledged that persistent inflammation in immune-mediated diseases, such as RA and psoriasis, have detrimental effects on the lymphatic system and chronic inflammatory conditions [9,10,22]. The effect of cytokines on lymphatic barrier function has been studied in vitro, but scRNA-seq data on transcriptional changes in LECs after exposure to cytokines have not yet been published [94]. More scRNA-seq studies to the effect of inflammation on transcriptome and function of LECs could help provide a new understanding of LECs’ contribution to the pathogenesis of chronic inflammatory diseases.

## 6. Concludings Remarks

In conclusion, scRNA-seq data accommodate exciting new insights in the embryonic development of LECs and the heterogeneity of LEC populations, and they provide novelties in terms of new functions and interactions with other cells. Furthermore, ScRNA-seq data helped in identifying more molecules involved in LEC and LN development. Research gaps regarding the origin of LECs in peripheral tissues could be further explored with the scRNA-seq techniques.

ScRNA-seq data, combined with other techniques, identified several LN LEC subpopulations with unique markers and functions, facilitating immune cells that travel to the LN to meet and activate each other. Floor LECs lining the SCS in the LN express many chemokines to attract various cell populations from peripheral tissues, and they rapidly upregulate chemokines in response to inflammation. Specialized LECs in the cortex of the LN regulate the subsequent process of retaining lymphocytes in the cortex to increase their chance to meet (dwelling time). Another function of the LN is to scan the lymph for lymph-borne pathogens. Furthermore, LN LEC subpopulations express molecules capable of enforcing effector cells to recognize antigens. The medullary sinus LECs even recruit neutrophils to battle specific lymph-borne pathogens in the medulla. Ceiling LECs lining the SCS of the LN also express molecules, such as *Cd36,* that can scavenge fatty acids from the lymph. Although future studies are needed to pinpoint some of these roles, the claim that LECs are indispensable for these LN functions is very clear. More research into inflamed LNs is needed, also to uncover novel molecules to target. 

Peripheral LEC populations include capillary LECs, collecting vessel LECs, and valve LECs. At a first glance, peripheral vessel LECs in most tissues and organs seem to express the same markers, such as LYVE-1, PDPN, VEGFR3, and PROX-1. Deeper sequencing depth and sequencing more LECs will help to find tissue- and spatial-specific differences between LECs. LECs are also in most tissues embedded between other stromal cells. Ablating LECs causes drastic transcriptomic changes in other cells, such as microglia and BECs, as was discovered in a meningeal LEC study. Vice versa, cells such as perivascular cells and fibroblasts are needed for proper LEC function and maintenance.

Although LECs are the corner stone of the host immune response, limited information on lymphatic vascular biology or the role of LECs in context of inflammation, in particular, have thus far been provided by technologies such as scRNA-seq studies. Currently, many chronic inflammatory diseases now rely on therapies targeted at immune cells. Increasing our knowledge on the role of LECs in chronic inflammation is needed to find novel therapeutic targets. However, targeting lymphatic function could influence both immune cells and nearby stromal cells, which could be a new powerful strategy. Modulating LEC function early in disease could even be a new preventive approach.

## Figures and Tables

**Figure 1 ijms-22-11976-f001:**
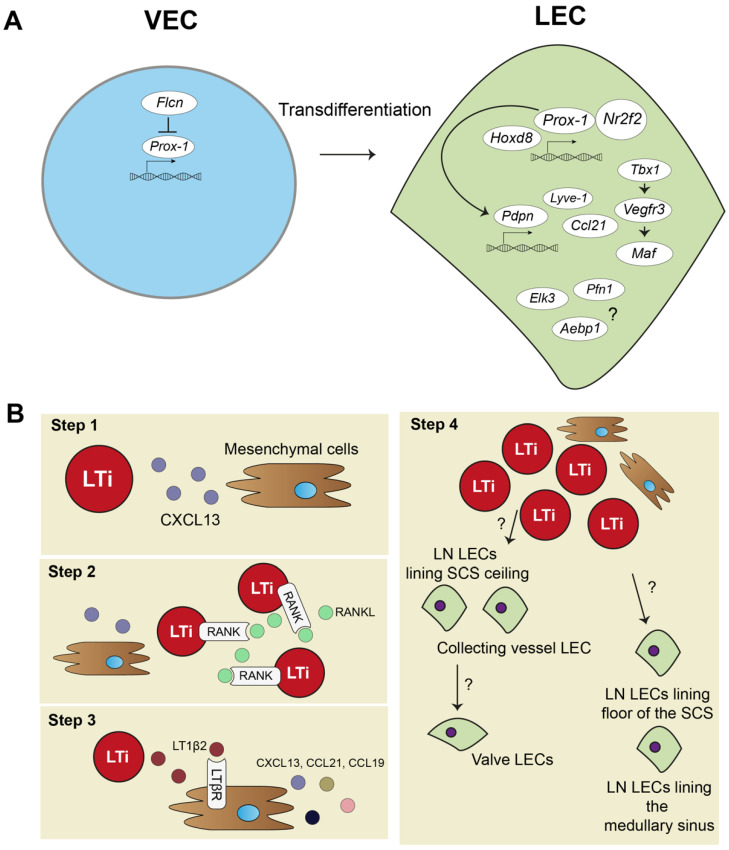
Lymphatic endothelial cell and lymph node embryonic development. (**A**) In embryonic development, lymphatic endothelial cells (LECs) mostly transdifferentiate from venous endothelial cells (VECs). Prospero homeobox protein 1 (Prox-1) is a master regulator of lymphatic fate. Homeobox d 8 (Hoxd8) can activate Prox-1 expression, and Prox-1 cooperates with Nuclear Receptor Subfamily2 Group F Member 2 (Nr2f2) to drive expression of other key peripheral LEC molecules, such as podoplanin(Pdpn), lymphatic vessel endothelial hyaluron receptor 1 (Lyve-1), chemokine (C-C motif) ligand 21 (Ccl21), and vascular endothelial growth factor receptor 3 (Vegfr3). Vegfr3 is also regulated by transcription factor T-box 1 (Tbx1) and can activate other transcription factors, such as Maf. Single cell RNA sequencing (scRNA-seq) data also suggested transcription factors, such as *ETS Transcription Factor 3 (Elk3*), or other protein coding RNAs, such as *Profilin 1* (*Pfn1*) and *AE binding protein 1* (Aebp1), to be involved in lymphatic development, but their function is not yet clear. (**B**) Lymph node (LN) development follows several steps: (1) mesenchymal cells secrete chemokine (C-X-C motif) ligand 13 (CXCL13) to attract lymphoid tissue inducer (LTi) cells. (2) LTi cells cluster together and activate receptor activator nuclear factor κ B (RANK) and RANK ligand (RANKL) signaling. (3) RANK-RANKL signaling increases lymphotoxin α1β2 (LT1β2), which activates lymphotoxin beta receptor (LTβR) on mesenchymal cells responding with more cytokine signaling and attracting more LTi cells. (4) Mesenchymal cells differentiate into stromal cell subsets of the LN, and LTi differentiate or attract LECs. Trajectory inference of scRNA-seq data showed that there are two developmental paths of the LN LECs.

**Figure 2 ijms-22-11976-f002:**
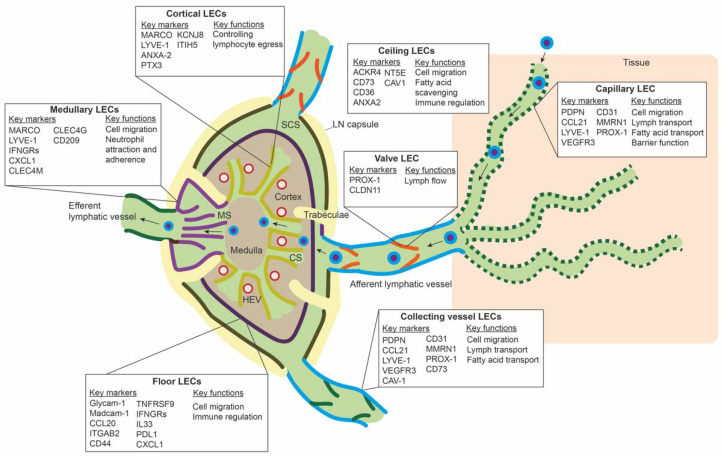
Overview of the different lymph node lymphatic endothelial cells and peripheral lymphatic endothelial cell subpopulations. The lymph node (LN) consists of multiple sinuses to transport lymph and immune cells from the afferent lymphatic vessels to the efferent lymphatic vessel. Immune cells arrive in the subcapular sinus (SCS) and migrate over the floor of the SCS, lined by floor LECs, to the LN cortex. The SCS is a narrow sinus, and it is close to the LN capsule. The LN capsule side of the SCS is lined by ceiling LECs. In the LN cortex, immune cells also arrive via high endothelial venules (HEVs). Egress of immune cells from the cortex is via the cortical sinuses (CS) lined by cortical LECs. These cortical sinuses drain in the medulla. Medullary sinuses (MS), lined by medullary LECs, transport lymph and immune cells to the efferent lymphatic vessel. LECs lining the different sinuses were found to be transcriptionally different and to have a range of different functions.

**Figure 3 ijms-22-11976-f003:**
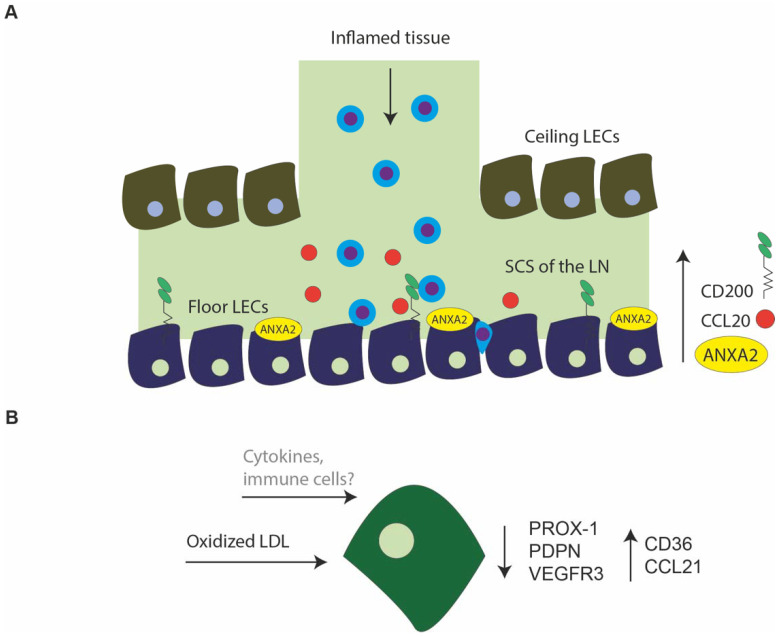
Inflammatory changes in lymph node lymphatic endothelial cells and peripheral lymphatic endothelial cells. (**A**) Predominantly, lymphatic endothelial cells (LECs) lining the floor of the subcapular sinus (SCS) (floor LECs) come into contact with immune cells of inflamed tissue. Floor LECs can rapidly upregulate the expression of chemokines, such as chemokine (C-C motif) ligand 20 (CCL20). Additionally, molecules such as Annexin A2 (ANXA-2) and cluster of differentiation 200 (CD200) are upregulated. The function of these molecules in the inflamed LN needs to be investigated. (**B**) Oxidized low-density lipoprotein (LDL) can downregulate transcripts like Prospero Homeobox Protein 1 (PROX-1), Podoplanin (PDPN), and Vascular Endothelial Growth Factor Receptor 3 (VEGFR3), which are key LEC markers. CD36, a fatty acid transporter, and CCL21 are upregulated. Single cell RNA sequencing (ScRNA-seq) data on inflammatory changes by cytokines or immune cells have not yet been investigated.

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
