# Peer review of "Single-Cell RNA Sequencing Reveals Heterogeneity and Functional Diversity of Lymphatic Endothelial Cells"

_ijms, 2021, doi:10.3390/ijms222111976_

Round 1

Reviewer 1 Report

In this review Braanker et al. described new discoveries in the field of lymphatic vascular biology (LEC development, heterogeneity and interactions with other cells) with a particular focus on those studies regarding single cell RNA sequencing technology. The topic is interesting but some minor questions have to be addressed.

  • Add some recent literature in the first part of the introduction, when speaking about the lymphatic vascular system
  • In the 2.1 paragraph, explain in few words the role of Hoxd8, Tbx1 and Elk3 as done with Maf. Are all of them transcription factors? Explain in the text as already done in the figure legend.
  • Do not use acronyms in the legend, write in full the acronyms the first time they appear.
  • The description of 2.2 Lymph node development is not too much clear. An additional image in figure 1, together with a better description of the text, could help in understanding.
  • Line 233 page 6, explain what MARCO is for
  • In the 5.1 paragraph please explain briefly the function of imiquimod.
  • Please check carefully some spelling errors in the abstract:

Line 13 – LECs have abilities to regulate

Line 24 - therapeutic targets

In figure 2 correct arive with arrive

Author Response

Dear reviewer 1,

Thank you for your response regarding our manuscript: “Single cell RNA sequencing reveals heterogeneity and functional diversity of lymphatic endothelial cells.”

We would like to thank the reviewer for their valuable comments to improve our manuscript. Based on these comments, we have made changes to the manuscript which are detailed in the point-by-point response below.

Point 1: Add some recent literature in the first part of the introduction, when speaking about the lymphatic vascular system

Response 1: Line 35 was adapted and 2 recent articles about the lymphatic vasculature were added.

Point 2: In the 2.1 paragraph, explain in few words the role of Hoxd8, Tbx1 and Elk3 as done with Maf. Are all of them transcription factors? Explain in the text as already done in the figure legend.

Response 2: Explanations were added according to reviewers’ suggestions.

Point 3: Do not use acronyms in the legend, write in full the acronyms the first time they appear.

Response 3: Acronyms in the legends are now written in full.

Point 4: The description of 2.2 Lymph node development is not too much clear. An additional image in figure 1, together with a better description of the text, could help in understanding.

Response 4: The text of 2.2 is adapted and figure 1B is modified to display the step-by-step process of lymph node development.

Point 5: Line 233 page 6, explain what MARCO is for

Response 5: Explanation for MARCO was added.

Point 6: In the 5.1 paragraph please explain briefly the function of imiquimod.

Response 6: A brief explanation of imiquimod was added.

Point 7: Please check carefully some spelling errors in the abstract: Line 13 – LECs have abilities to regulate, Line 24 - therapeutic targets

Response 7: Spelling errors were corrected.

Point 8: In figure 2 correct arive with arrive

Response 8: Spelling error was corrected.

Reviewer 2 Report

The manuscript “Single cell RNA sequencing reveals heterogeneity and functional diversity of lymphatic endothelial cells” by den Braanker et al. gathers together the more recent findings on lymphatic endothelial cells (LEC) diversity obtained with scRNA-seq.

Using scRNA-seq publications, the authors focus first on embryonic development and the new potential genes involved in embryonic lymphatic development. Then, the review emphasises on the heterogeneity of LEC, notably within the lymph nodes and how the LEC interact with other stromal cells. Finally, the authors underline the changes observed in LEC following inflammation.

Strengths: this review includes a comprehensive and complete literature revision on LEC identity using the scRNA-seq publications. Figure 2 allow a quick readout of the LEC heterogeneity by highlighting specific key markers and key functions.

Limitations: although the aim of the review is to focus on recent scRNA-seq data, it would have been appropriate to mention some studies supporting LEC heterogeneity from different organs/localisation using other techniques (e.g. Garrafa et al. Lymphology, 2005; C Normen et al. Blood, 2010; Mouta-Bellum et al. Dev Dyn, 2009). In addition, and as the authors mentioned line 77, recent studies have also shown non-venous sources for LEC during development (e.g. Stanczuk et al. Cell Reports, 2015; Klotz et al. Nature, 2015) and a brief description in the part “2.1. Peripheral lymphatic vessel formation” would have been advantageous.

Author Response

Dear reviewer 2,

Thank you for your response regarding our manuscript: “Single cell RNA sequencing reveals heterogeneity and functional diversity of lymphatic endothelial cells.”

We would like to thank the reviewer for their valuable comments to improve our manuscript. Based on these comments, we have made changes to the manuscript which are detailed in the point-by-point response below.

Point 1: It would have been appropriate to mention some studies supporting LEC heterogeneity from different organs/localisation using other techniques (e.g. Garrafa et al. Lymphology, 2005; C Normen et al. Blood, 2010; Mouta-Bellum et al. Dev Dyn, 2009).

Response 1: We mentioned the proposed studies.

Point 2: In addition, and as the authors mentioned line 77, recent studies have also shown non-venous sources for LEC during development (e.g. Stanczuk et al. Cell Reports, 2015; Klotz et al. Nature, 2015) and a brief description in the part “2.1. Peripheral lymphatic vessel formation” would have been advantageous.

Response 2: We mentioned the proposed studies and adapted the start of paragraph 2.1.